

# Contrasting two major Arctic coastal polynyas: the role of sea ice in driving diel vertical migrations of zooplankton in the Laptev and Beaufort Seas

Igor A. Dmitrenko[1*], Vladislav Petrusevich[2], Andreas Preußer[3], Ksenia Kosobokova[4], Caroline Bouchard[5,6], Maxime Geoffroy[7,8], Alexander S. Komarov[9], David G. Babb[1], Sergei A. Kirillov[1], and David G. Barber[1†]

[1]Centre for Earth Observation Science, University of Manitoba, Winnipeg, R3T 2N2, Manitoba, Canada
[2]Northwest Atlantic Fisheries Centre, Fisheries and Oceans Canada, St. John's, A1A 5J7, Newfoundland and Labrador, Canada
[3]Alfred Wegener Institute for Polar and Marine Research, Bremerhaven, 27570, Germany
[4]Shirshov Institute of Oceanology, Russian Academy of Sciences, Moscow, 117218, Russia
[5]Greenland Climate Research Centre, Greenland Institute of Natural Resources, Nuuk, 3900, Greenland
[6]Département de Biologie, Université Laval, Québec, G1V 0A6, Canada
[7]Centre for Fisheries Ecosystems Research, Fisheries and Marine Institute, Memorial University of Newfoundland, St. John's, A1C 5R3, Newfoundland and Labrador, Canada
[8]Department of Arctic and Marine Biology, Arctic University of Norway, Tromsø, 9019, Norway
[9]Data Assimilation and Satellite Meteorology Research Section, Environment and Climate Change Canada, Ottawa, K1A 0H3, Ontario, Canada
†David G. Barber passed away on 15 April 2022

*Correspondence to*: Igor A. Dmitrenko (igor.dmitrenko@umanitoba.ca)

**Abstract.** The diel vertical migration (DVM) of zooplankton is one of the largest species migrations to occur globally and is a key driver of regional ecosystems and the marine carbon pump. Previously thought to be hampered by the extreme light regime prevailing in the Arctic Ocean, observations have revealed that DVM does occur in ice-covered Arctic waters and suggest the decline in Arctic sea ice may thereby impact DVM and its role in the Arctic ecosystem. However, coastal polynyas present a unique environment where open water or thin, nearly translucent, ice prevail when offshore winds advect the ice pack away from the coast, allowing light into the surface waters and potentially disrupting DVM. Here, four yearlong time series of acoustic backscatter collected by moored acoustic Doppler current profilers at two opposite sides of the circumpolar polynya system at the Laptev Sea shelf (2007-08) and the Beaufort Sea shelf (2005-06) were used to examine the annual cycle of acoustic scattering, and therefore the annual cycle of DVM in these areas. The acoustic time series were used along with atmospheric and oceanic reanalysis and satellite data to interpret the results. Our observations show that DVM started to occur once the ice-free surface or under-ice layer irradiance exceeds a certain threshold (from ~0.3 to 3.3 lux), which is about two to ten times lower in the Beaufort Sea compared to the Laptev Sea. In the Laptev Sea, DVM does not occur during polar night, while civil twilight in the Beaufort Sea is sufficient to trigger DVM through polar night. This difference in DVM between the Laptev and Beaufort Seas is not entirely assigned to the 3° difference in latitude between the mooring positions, but also to the different light threshold required to trigger DVM, different zooplankton communities'





composition, and potentially different depths and predation pressure. We find examples in both the Laptev and Beaufort Seas where the formation of polynyas and large leads caused DVM to abruptly cease or be disrupted, which we attribute to predator avoidance by the zooplankton in response to higher polar cod abundance near the open water. Finally, light attenuation by sea-ice in the Beaufort Sea caused DVM to extend onto the polar day until summer solstice. Overall, our

results highlight the role of sea ice in disrupting synchronized DVM, the spatial variability in the relationship between sea ice and DVM, and the potential ecological impact of significant trends toward a more extensive circumpolar Arctic coastal polynya as part of changing ice conditions in the Arctic Ocean.

## 1 Introduction

Every day zooplankton migrates vertically to rise at sunset and to descent with sunrise according to the daily cycle of

irradiance. This process is known as diel vertical migration (DVM). DVM is a geographically and taxonomically widespread behavior of zooplankton that spans across diel and seasonal timescales following the latitudinal variation in daylight (Bandara et al., 2021). DVM is arguably the most widespread daily migration of animals on Earth (Hays, 2003) and the largest non-human migration (Brierley, 2014). For the mesopelagic twilight zone, the biological pump and the biogeochemistry is significantly impacted by DVM (e.g., Archibald et al., 2019). The carbon transported by DVM can

account for up to 30% of the total particulate flux and as much as 50% of the metabolic carbon dioxide produced in the twilight zone (Bianchi et al., 2013).

In all oceans, including the Arctic Ocean, DVM is primarily driven by irradiance (e.g., Hobbs et al., 2021). However, over the past two decades DVM was observed to occur during the polar night (Berge et al., 2009, 2015), and more recent studies have analyzed the role of moonlight in modifying DVM during polar night (Last et al., 2016; Petrusevich et al., 2016).

Furthermore, oceanographic research using moored acoustic Doppler current profilers (ADCPs) revealed the role of wind-driven and tidal-driven water dynamics in modifying DVM (Petrusevich et al., 2016, 2020; Dmitrenko et al., 2020).

In the Arctic Ocean, the ice cover attenuates light transmission and therefore modifies DVM (Hobbs et al., 2018, 2021; Dmitrenko et al., 2020; Flores et al., 2023). Thereby, over the Arctic and sub-Arctic shelves, seasonally covered with perennial sea-ice, DVM is also partially controlled by the ice cover (Wallace et al., 2010; Dmitrenko et al., 2021). For this

reason, one may assume that small- and large-scale disruptions of the sea-ice cover could impact DVM locally and regionally. A thinner sea ice cover in the central Arctic Ocean could thus keep zooplankton deeper for longer (Flores et al., 2023). The seasonal variations in irradiance and ice cover also play a role in triggering the zooplankton seasonal vertical migration (SVM) over the Arctic and sub-Arctic shelves. Beyond irradiance, SVM is also driven by adaptation to seasonality in resources. In high latitudes, an upward migration of herbivorous zooplankton in spring utilizes the high primary

production in surface layers during spring bloom, followed by a downward migration in autumn to reside at greater depth during the non-productive autumn and winter (Conover and Huntley, 1991; Conover and Siferd, 1993; Søreide et al., 2022).



Beyond light, the sea-ice disruptions can deviate zooplankton vertical migrations through the polynya-related processes. The persistent regions of open water and thin ice in the Arctic Ocean and the adjoining Arctic shelves called polynyas substantially impact physical and biogeochemical processes in the water column. Polynyas provide access to light. In combination with resuspension of nutrients, this extends the normally short Arctic phytoplankton bloom (Tremblay et al., 2002; Arrigo and van Dijken, 2004; Elias, 2020), attracting zooplankton and fish, providing a rich harvest for marine predators and access to birds and mammals in the otherwise ice-covered water (Arrigo, 2007; Heide-Jørgensen et al., 2013). Over the Arctic shelves, the circumpolar polynya system forms off fixed obstacles comprised by unmovable landfast ice area under offshore wind forcing. Following the inventory of Barber and Massom (2007), north of 68°N, the circumpolar Arctic polynya is comprised of 17 individual polynya regions. They are described in detail in Preußer et al. (2016, 2019). Among these 17 polynya regions, the Laptev Sea polynya of the Siberian shelf and the Cape Bathurst polynya of the Beaufort Sea shelf (Figure 1) appear to be rather comparable in terms of polynya area and accumulated sea-ice production. On average, from December to March (2002/2003 to 2017/2018), the Laptev Sea polynya extends over $10.6 \pm 4.2$ km2 generating $70 \pm 28$ km3 of new ice (Preußer et al., 2019). The Cape Bathurst polynya is marginally smaller, for the same period extending to $7.5 \pm 4.4$ km2 and generating $46 \pm 26$ km3 of new ice (Preußer et al., 2019). The Laptev Sea polynya was extensively studied in 2007–2012 under the framework of the Russian-German project named "The Eurasian Shelf Seas in the Arctic's Changing Environment: Frontal Zones and Polynya Systems in the Laptev Sea" (Kassens, 2012) and during subsequent projects from 2013 to 2021. The Cape Bathurst polynya was assessed in 2007-08 during the Canadian project named "The Circumpolar Flaw Lead (CFL) system study" (Barber et al., 2010). These two projects provided data and scientific background for assessing the role of polynyas in deviating or even disrupting the DVM seasonal cycle, which is a focus of the present study. In what follows, we specifically focus on the southeastern Laptev Sea shelf in the eastern (Russian) Arctic and the southeastern Beaufort Sea shelf in the western (Canadian) Arctic. We evaluate the DVM and its modification following the formation of a polynya traced by satellites. Moreover, we quantitatively assess the intensity of acoustic backscatter during polar day and polar night in response to solar irradiation, putting its spatial variability in the context of atmospheric forcing and sea-ice conditions.



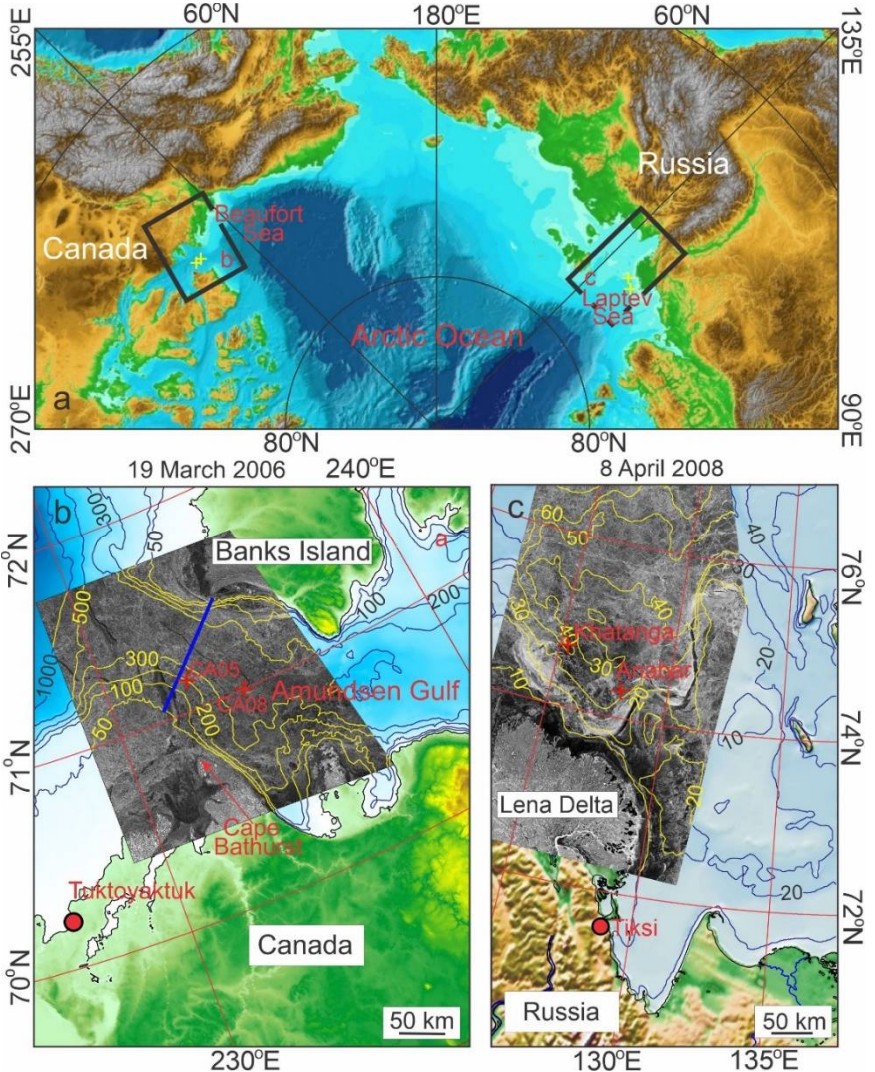

**Figure 1:** (**a**) Map of the eastern Arctic Ocean adopted from *Jakobsson et al.* (2000) with positions of the year-long oceanographic moorings (yellow crosses) deployed in the Beaufort and Laptev Seas. Black rectangles show the eastern Beaufort and Laptev Seas, enlarged in **b** and **c**, respectively. (**b**) Map of the eastern Beaufort Sea extended to the Amundsen Gulf with overlaid RADARSAT-1 satellite image from 19 March 2006. Deep blue, white and yellow lines depict bathymetric contours. Blue line shows CTD section sampled in September 2005 and 2006. Red crosses mark moorings CA05-05 and CA08-05 deployed from September 2005 to September 2006 in the area of Cape Bathurst polynya. (**c**) Map of the southeastern Laptev Sea shelf with overlaid Envisat advanced synthetic aperture radar (ASAR) image from 8 April 2008. Deep blue and yellow lines depict bathymetric contours. Red crosses mark moorings Khatanga and Anabar deployed from September 2007 to September 2008 in the area of Siberian coastal polynya.



## 2 Materials and Methods

### 2.1 Study Area

The Laptev Sea shelf represents one of the shallowest and broadest shelf regions of the entire World Ocean, extending a distance of ~500 km offshore (Figures 1a and 1c). It is strongly impacted by river runoff. The study area in the eastern Laptev Sea shelf (Figure 1c) receives an average of 535 km$^3$ of freshwater from the Lena River annually with ~35% of the annual discharge occurring in June (Yang et al., 2002). The distribution of the riverine water over the Laptev Sea shelf is entirely wind driven. The cyclonic atmospheric forcing diverges the river plume alongshore eastward, while the anticyclonic wind moves the riverine water offshore northward (Dmitrenko et al., 2005a, 2008, and 2010b). The Laptev Sea shelf is seasonally ice-covered from about October to July (Onarheim et al., 2018). Most of the shelf area is covered by landfast sea ice extended approximately to the 20 m depth contour, while mobile first-year pack ice drifts beyond the landfast ice edge. The fast ice reaches a peak thickness in April between 1.5 m further offshore (Belter et al., 2020) and about 2.0 m at the front of the Lena Delta (Hendricks et al., 2018). The interface of the fast ice and pack ice is dynamic; it is frequently covered by a vast latent heat polynya that forms when offshore winds advect the mobile pack ice away from the landfast ice edge (e.g., Dmitrenko et al., 2005b; Willmes et al., 2010, 2011; Kirillov et al., 2013; Gutjahr et al., 2016). The mean ice thickness within the polynya is ~11 cm, with approximately half of the polynya covered by ice with thickness of 12–20 cm, and less than 2% of the polynya covered by open water and ice thinner than 2 cm (Willmes et al., 2011). The polynyas impact on sea-ice production and vertical mixing in the Laptev Sea has been extensively studied (e.g., Dmitrenko et al., 2008, 2009, 2010a,b, 2012; Willmes et al., 2010, 2011; Kirillov et al., 2013; Preußer et al., 2016, 2019).

The zooplankton communities of the Laptev Sea shelf are composed mostly of copepods, with a domination of small-sized (1.0–1.5 mm body length) organisms (Abramova and Tuschling, 2005; Arashkevich et al., 2018). In the coastal regions of the eastern and south-eastern Laptev Sea, small-sized brackish-water neritic species tolerant to low salinity dominate (Kosobokova et al., 1998; Pasternak et al., 2022). The western and northeastern Laptev Sea affected by advection of the oceanic water hosts larger-sized (2.0–4.5 mm body length) marine species (*Calanus* spp., *Metridia* longa) in addition to small brackish-water taxa. In summer, the larger species are present in the regions with depth >50 m (Kosobokova et al., 1998), but by winter they migrate to the deeper regions of the Laptev Sea continental slope. DVM of zooplankton in the shallow Laptev Sea seems to be conducted by copepods *Pseudocalanus* spp., *Acartia* spp., and mysids (Dmitrenko et al., 2021), while in the deeper regions (>50 m depth) *Calanus* spp. and *M. longa* are known to perform DVM in summer (Arashkevich et al., 2018). The impact of the Laptev Sea polynya on DVM has not been sufficiently assessed. Only recently Dmitrenko et al. (2021) revealed that the DVM disruptions were attributed to polynya opening. However, for the shallow and strongly stratified water column of the Laptev Sea, the comprehensive analysis of the polynya impact on DVM requires further evaluation.





In contrast to the Laptev Sea shelf, the Beaufort Sea   has a narrow continental shelf <110 km wide with waters as deep as 3800 m farther offshore in the Canada Basin (Sharma, 1979; Carmack and MacDonald, 2002; Figures 1a and 1b). The Mackenzie River, Canada's longest and largest river, provides the greatest Western Hemisphere discharge to the Arctic
Ocean (Rood et al., 2017) transporting an average of 325 km$^3$ of freshwater per year through the Beaufort Sea shelf (Yang et al., 2015). During May–June, the Mackenzie River flow exceeds that for January–April by about four times (Yang et al., 2015). Wind forcing controls the diversion of the river plume over the Beaufort Sea shelf (Mulligan et al., 2010; Mol et al., 2018; Mulligan and Perrie, 2019). The along-shore easterly (anticyclonic) wind diverges the Mackenzie River riverine water westward, and also favours upwelling of the Pacific- and Atlantic-derived Arctic water onto the Beaufort Sea shelf. In
contrast, the along-shore westerly (cyclonic) wind forces eastward transport of the Mackenzie River plume, and generates downwelling and storm surges along the Beaufort Sea coast (Carmack and Macdonald, 2002; Dmitrenko et al., 2016, 2018). Freeze-up in the eastern Beaufort Sea begins in October/November, and break-up begins in May (Smith and Rigby 1981; Johnson and Eicken, 2016; Onarheim et al., 2018). The Cape Bathurst polynya forms in the southeastern Beaufort Sea between Cape Bathurst and Banks Island extending to the Amundsen Gulf (Stirling, 1980; Smith and Rigby, 1981; Barber
and Hanesiak, 2004; Figure 1b). The polynya can be open any time when an ice cover is present, but starts becoming prominent in April (Smith and Rigby, 1981). Landfast ice in this area extends offshore to about the 20 m isobath beyond which the polynya is bounded to the north by drifting pack involved to the Beaufort Sea Gyre (Carmack and MacDonald, 2002). In general, the Cape Bathurst polynya is a consequence of the Beaufort Sea Gyre acting like an ice bridge (Barber and Hanesiak, 2004) and creating conditions conducive to oceanic upwelling recurrently observed in the area (Williams and
Carmack, 2008; Sévigny  al., 2015).

In the Beaufort Sea, both zooplankton and polar cod conduct DVM from August until May, but the synchronized DVM stops during polar day (Geoffroy et al. 2016; Dimitrenko et al. 2020). The zooplankton communities of the Beaufort Sea are characterized by different species composition and quantitative structure compared to the communities of the shallow Laptev Sea shelf. In contrast to the Laptev Sea, the larger-sized Arctic-oceanic taxa prevail in the Beaufort Sea (Darnis et al., 2014,
2022). Among them the most active diel migrators are the copepods *Metridia longa*, late copepodites of *Calanus glacialis* and chaetognaths *Parasagitta elegans*, all falling within the size range 2.0–20 mm, which is well detectable by acoustic backscatter. Most copepods also conduct SVM, with large herbivorous copepods such as *Calanus hyperboreus* and *Calanus glacialis* entering diapause and descending to the deep Atlantic waters for overwintering and ascending from depths to the surface layers in spring. The smaller zooplankton, the omnivorous copepods Oithona similis and Pseudocalanus spp., occupy
intermediate depths continuing to feed and, presumably, to reproduce during the winter season (Darnis and Fortier, 2012; 2014). Most fish and zooplankton avoid the colder Polar Mixed Layer at the surface year-round (Geoffroy et al. 2011; Darnis and Fortier, 2014). Polar cod also perform SVM towards deep embayment during the winter, presumable for spawning (Benoit et al., 2010; Geoffroy et al., 2011). To our knowledge, the impact of the Cape Bathurst polynya on DVM of fish and zooplankton was never assessed.



Overall, among all DVM studies in the Arctic shelves, only Dmitrenko et al. (2021) evaluated the role of coastal polynyas in disrupting DVM over the Laptev Sea shelf. They suggested a predator-avoidance behavior of zooplankton conditioned by higher polar cod abundance attracted by the polynya opening. To date there has yet to be a comprehensive analysis of how DVM is modified by the opening of a polynya, and what is a regional difference of the DVM response to polynya formation at the eastern and western Arctic shelves.

**2.2 Observations and Data**

Here we provide analysis of DVM at two opposite sides of the circumpolar polynya system (Figure 1a) at the eastern Beaufort Sea shelf (Canadian Arctic; Figure 1b) and the Laptev Sea shelf (Russian Arctic; Figure 1c) and give examples of how the opening of a polynya and/or leads affects DVM. Observing DVM in ice-covered waters is difficult; however, moored acoustic Doppler current profilers (ADCPs) provide a time series of acoustic backscatter in the water column that

can be used to trace DVM year-round. Oceanographic moorings, equipped with ADCPs, have been deployed annually over the Laptev and eastern Beaufort Sea shelves since the end of 1990s. While ADCPs provided valuable information on physical oceanography, this data set was rarely assessed to analyze biological backscatter and vertical migrations of zooplankton. Here, we use the ADCP-derived time series of acoustic backscatter from two year-long mooring arrays (Table 1) deployed in the southeastern Beaufort Sea in September 2005 (moorings CA05 and CA08; Figure 1b) and in the

southeastern Laptev Sea in September 2007 (moorings Anabar and Khatanga; Figure 1c) to examine polynya impact on DVM. All four moorings were located above the Arctic Circle at 66.6 °N (Table 1) at about 74.5°N (Laptev Sea) and 71°N (the Beaufort Sea), the area where the sun is between 6° and 12° below the horizon all day on the winter solstice. At these latitudes, during the nautical polar night period, there is no trace of daylight, and only around midday there is slight light because of refraction, and the horizon and the brighter stars are visible during clear sky. During polar day around the summer

solstice, the sun does not move below the horizon, and daylight lasts from sunset to sunrise.

Observations of DVM were derived from four up-looking ADCPs deployed on bottom anchored oceanographic moorings designed as an anchored taut line with subsurface flotations and acoustic releases. More details on these moorings can be found in Table 1, and ADCP data are available in Dmitrenko (2024). The ADCPs were 300 kHz upward-looking Workhorse Sentinel ADCPs from Teledyne RD Instruments. For this study, we used only the acoustic backscatter intensity data that was

obtained at intervals of 1 m in the Laptev Sea and 8 m in the Beaufort Sea, with a 30 min and 20 min ensemble time interval and 30 and 20 pings per ensemble for the Laptev and Beaufort Seas, respectively. ADCPs sampled the overlying water column except the shadow zone over the acoustic transducer and the sub-surface water layer where the acoustic reflection from the air-water and/or ice-water interface compromises the data (Table 1). In the shallow Laptev Sea, nearly the entire water column has been sampled, while in the Beaufort Sea, ADCPs sampled only the upper portion of the water column

from 9-10 m to 65 and 82 m depth for CA05 and CA08, respectively (Table 1).




The ADCP-derived vertical velocity has also been used to study rates of zooplankton DVM in the Beaufort Sea (e.g., Plueddemann and Pinkel, 1989; Cottier et al., 2006; Ochoa et al., 2013). The accuracy of the ADCP vertical velocity measurements is not validated; however, the manufacturer reports that the vertical velocity is more accurate, by at least a factor of 2, than the horizontal velocity (Wood and Gartner, 2010) and has an accuracy of ±1.0%. Nevertheless, vertical
velocities in the 3 mm s$^{-1}$ range may well be within the noise level (Miller, 2003).

**Table 1**: Description of ADCP moorings in the Beaufort and Laptev Seas

| Mooring | Lat. N | Lon. E | Depth (m) | ADCP depth (m) | Valid ADCP data from | To | Bin size (m) | Depth range from/to (m) |
|---|---|---|---|---|---|---|---|---|
| Beaufort Sea (2005-2006) | | | | | | | | |
| CA05-05 | 71° 16.8’ | 232° 27.8’ | 201 | 90 | 9 Sep 2005 | 11 Jul 2006 | 8 | 10/82 |
| CA08-05 | 71° 00.4’ | 233° 55.5’ | 397 | 73 | 10 Sep 2005 | 18 Jul 2006 | 8 | 9/65 |
| Laptev Sea (2007-2008) | | | | | | | | |
| Anabar | 74° 19.9’ | 128° 00.0’ | 32 | 30 | 2 Sep 2007 | 7 Sep 2008 | 1 | 3/27 |
| Khatanga | 74° 42.9’ | 125° 17.4’ | 43 | 40 | 3 Sep 2007 | 8 Sep 2008 | 1 | 7/40 |

Fields of sea level pressure (SLP; Figure S1), 10-m wind velocity, and 2-m air temperature at 6-h intervals (Figure S2) were retrieved from the ERA5 atmospheric reanalysis (Copernicus Climate Change Service, 2017; Hersbach et al., 2020). The
horizontal resolution of ERA5 is 31 km. The daily mean cloud fraction data (the percentage of Earth's surface covered by clouds) were derived from the Aqua Moderate Resolution Imaging Spectroradiometer (MODIS) satellite imagery (Platnick et al., 2017). The sensor/algorithm resolution is 5 km, imagery resolution is 2 km, and the temporal resolution is daily. For both the ERA5 and cloud fraction we used data from the grid point closest to the mooring positions in our analysis.

The extent and evolution of the sea-ice cover around the moorings between September 2005 and July 2006 in the Beaufort
Sea and between September 2007 and September 2008 in the Laptev Sea were analyzed in synthetic aperture radar (SAR) images collected by RADARSAT-1 and Envisat, respectively. Within the radar images, areas of higher radar backscatter (gray and white in Figures 1b and 1c) indicate the formation of new ice in the polynya, while lower radar backscatter (black and gray) indicates open water within the coastal polynya and leads in the surrounding ice pack. The fast ice edge separates the landfast ice from the coastal polynya and is usually evident in the radar images (e.g., Figure 1c). For cloud free
conditions, when the sun is above the horizon, we also used MODIS true-color satellite imagery from NASA (https://modis.gsfc.nasa.gov; Figures 2 and 3). MODIS imagery spatial resolution is 250 m, and the temporal resolution is daily.





Monthly gridded fields of sea ice concentration around the mooring positions were retrieved from the National Snow and Ice
Data Center (Walsh et al., 2019). The ice concentration dataset at 25 km resolution was produced by the NASA Team
algorithm (Comiso et al., 1997) applied to passive microwave brightness temperatures (Cavalieri et al., 1996). Observations
of snow depth are from the Advanced Microwave Scanning Radiometer - Earth Observing System (AMSR-E) at 12.5 km
spatial resolution (Cavalieri et al., 2014). Snow depth over sea ice is a running 5-day average based on the current day and
the 4 previous days.

**Figure 2:** The MODIS true-color satellite imagery for February to August 2008 shows the evolution of ice conditions in a
relatively cloud-free area of Siberian coastal polynya in the vicinity of the Lena Delta (Laptev Sea). Red crosses mark
moorings Khatanga and Anabar. Blue crosses depict the spatial node where 2-m air temperature and 10-m wind were derived
from ERA5 (Figures S2a and S2b, respectively). Red arrows show daily mean wind for the day the image was taken.



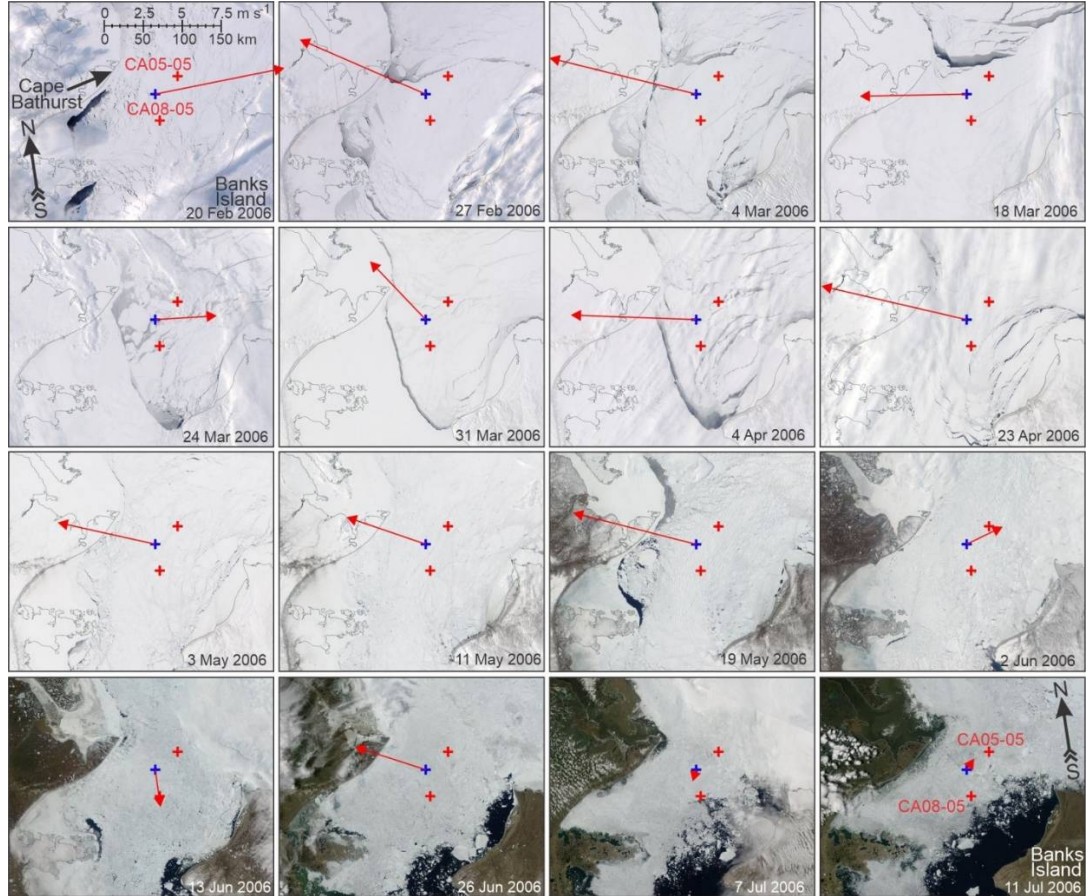

**Figure 3**: The MODIS true-color satellite imagery for February to July 2006 shows the evolution of ice conditions in a relatively cloud-free area of Cape Bathurst polynya in the Amundsen Gulf (Beaufort Sea). Red crosses mark moorings CA05-05 and CA08-05. Blue crosses depict the spatial node where 2-m air temperature and 10-m wind were derived from ERA5 (Figures S2c and S2d, respectively). Red arrows show daily mean wind for the day the image was taken.

During the time of mooring deployment, the spatial and temporal coverage of sea-ice thickness observations from spaceborne satellite altimeters was insufficient. Hence, in our analysis we use modeled daily estimates of sea-ice thickness around the mooring positions from the Pan-Arctic Ice Ocean Modeling and Assimilation System (PIOMAS; Figures 4–7a). PIOMAS is a coupled ocean and sea ice model that assimilates daily sea ice concentration and sea surface temperature satellite products (Zhang and Rothrock, 2003). In general, PIOMAS tends to overestimate the thickness of thin ice and underestimate the thickness of thick ice (Schweiger et al., 2011). For the eastern Laptev Sea shelf, from 2003 to 2008 PIOMAS overestimated the thickness of ice relative to ICESat observations by 0.2–0.7 m (Schweiger et al., 2011). This uncertainty can generate significant bias in estimated under-ice irradiance during winter. To better constrain the uncertainty in ice thickness, we use a record of daily mean sea ice draft derived from the bottom tracking mode of the ADCPs deployed



in the Laptev Sea (Belter et al., 2020). However, this approach also has its uncertainties, as comparing ice drafts derived by

ADCP and up-ward Looking Sonars (ULS) reveals the ADCPs underestimate the mean and median thickness by approximately 0.10 to 0.37 m (Figures 4a and 5a).

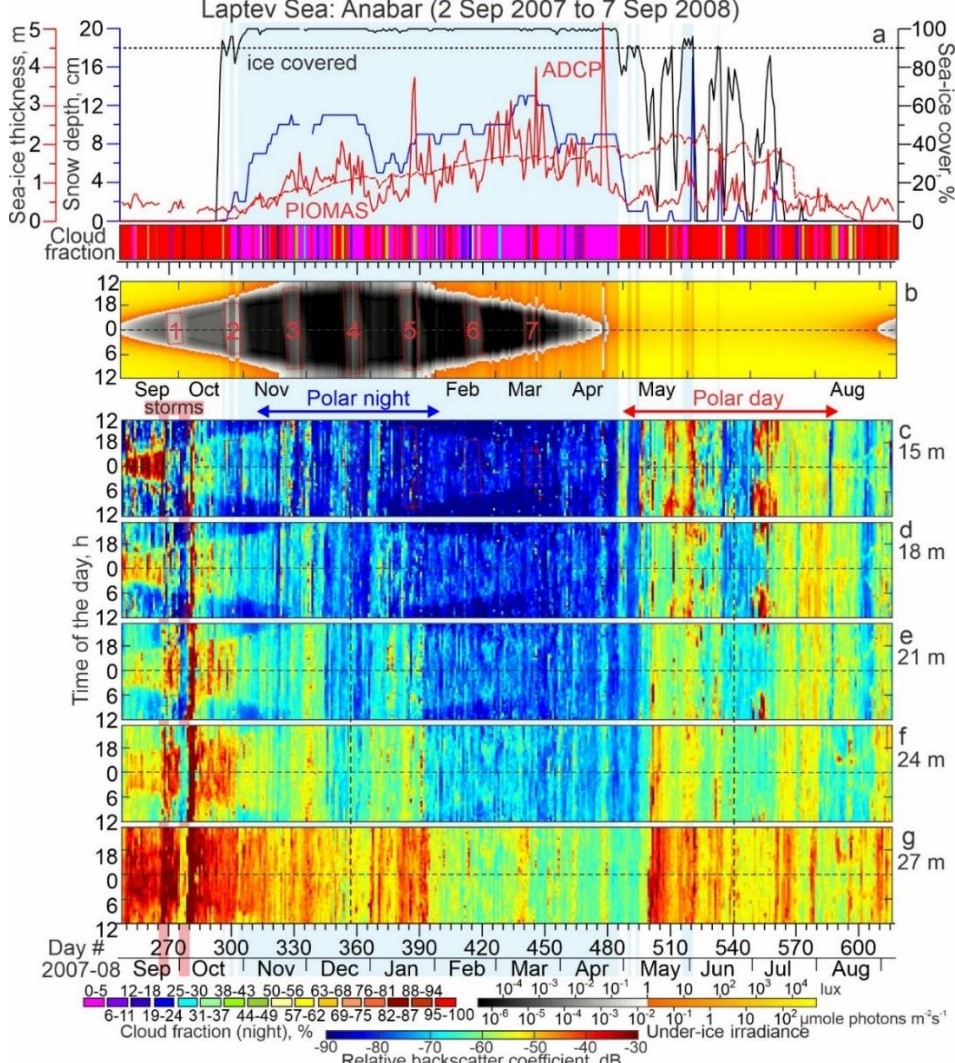

**Figure 4**: Observations at Anabar. Blue shading highlights sea-ice cover exceeding 90%. (**a**) Time series of the snow depth (cm, blue), sea-ice thickness (m, red) and concentration (%, black), and nighttime cloud fraction (shown in color, %). Sea-ice

thickness is from ADCP (solid red line) and PIOMAS (dashed red line). Actograms of the (**b**) under-ice irradiance (lux) and (**c-g**) mean volume backscatter strength (MVBS; dB) at five depth levels: (**c**) 15, (**d**) 18, (**e**) 21, (**f**) 24, and (**g**) 27 m depth. (**b**) Red dotted rectangles with reference numbers depict the full moon occurrences. (**c-g**) Red and blue arrows at the top indicate the periods of polar day and polar night, respectively. Black vertical lines depict solstices. Pink shading highlights two storms following Hölemann et al. (2011).




**Figure 5**: Observations at Khatanga. Blue shading highlights sea-ice cover exceeding 90%. (**a**) Time series of the snow depth (cm, blue), sea-ice thickness (m, red) and concentration (%, black), and nighttime cloud fraction (shown in color, %). Actograms of the (**b**) under-ice irradiance (lux) and (**c-g**) mean volume backscatter strength (MVBS; dB) at five depth levels: (**c**) 15, (**d**) 20, (**e**) 25, (**f**) 30, and (**g**) 35 m depth. All other designations are similar to those in Figure 4.





**Figure 6**: Observations at CA05. Blue shading highlights sea-ice cover exceeding 90%. (**a**) Time series of the snow depth (cm, blue), sea-ice thickness (m, red) and concentration (%, black), and nighttime cloud fraction (shown in color, %). Sea-ice thickness is from PIOMAS (red line). Actograms of the (**b**) under-ice irradiance (lux) and (**c-g**) mean volume backscatter strength (MVBS; dB) at five depth levels: (**c**) 15, (**d**) 31, (**e**) 47, (**f**) 55, and (**g**) 63 m depth. Pink shading highlights the lead event in December 2005 and March 2006 (see Figures 13 and 3, respectively). All other designations are similar to those in Figure 4.







**Figure 7**: Observations at CA08. Blue shading highlights sea-ice cover exceeding 90%. (**a**) Time series of the snow depth (cm, blue), sea-ice thickness (m, red) and concentration (%, black), and nighttime cloud fraction (shown in color, %). Sea-ice thickness is from PIOMAS (red line). Actograms of the (**b**) under-ice irradiance (lux) and (**c-g**) mean volume backscatter strength (MVBS; dB) at five depth levels: (**c**) 15, (**d**) 31, (**e**) 47, (**f**) 55, and (**g**) 63 m depth. Pink shading highlights the lead events in November 2005 and January 2006 (see Figure 13). All other designations are similar to those in Figure 4.



For a more precise analysis of the impacts of polynyas and leads on DVM, we used a lead dataset retrieved from MODIS thermal infrared imagery. Reiser et al. (2020) use ice surface temperatures (IST) swath data to retrieve a lead data product that features pan-Arctic and quasi-daily categorical maps of leads (with accompanying lead-score), artefacts, clouds and sea-ice for the period from 2002 to 2020 (November to April). Thereby, dedicated image-processing strategies yield multiple lead metrics that are then utilized to filter out unidentified cloud-artefacts in a fuzzy logic approach (Willmes et al., 2015, 2016). Preußer et al. (2022) indicated that these classified leads often comprise ice thicknesses beyond 20 cm, a commonly used thickness threshold for polynyas and leads. In order to derive parameters that describe the lead activity in proximity of the mooring locations, the MxD29 (MxD: MOD29 from Terra and MYD29 from Aqua) Collection 6 sea-ice product (Hall et al., 2004; Riggs et al., 2015) served as the fundamental data basis. It contains IST calculated from MODIS thermal infrared satellite data at a nominal geometric resolution of 1 km² at nadir. The MODIS cloud mask (MxD35, Ackerman et al., 2010) acts as a first cloud filter as part of the MxD29 product, with known difficulties to detect low clouds and sea smoke during winter.

Complimentary conductivity-temperature-depth (CTD) profiles were collected at each mooring location during deployment and recovery (Figure S3) as well as over the entire area at the front of the Lena Delta in the Laptev Sea (Transdrift, 2007, 2008; Figure S4) and across the entrance to Amundsen Gulf in the Beaufort Sea (Gratton et al., 2014a, 2014b; Figure S5). In the Laptev Sea, CTD profiles were collected with a CTD probe SBE 19plus, while in the Beaufort Sea, a CTD probe SBE 911plus was used. According to the manufacturer estimates, individual conductivity measurements are accurate to 0.0003 and 0.0005 Sm$^{-1}$, respectively, for the SBE 911plus and SBE 19plus. Individual temperature measurements are accurate to 0.001 and 0.005 °C, respectively. The CTD instruments were calibrated by the manufacturer before each expedition.

### 2.3 Methods

The methodology used in this study is largely based on that applied by Petrusevich et al. (2016, 2020) and Dmitrenko et al. (2020, 2021). We analyzed the year-long time series of acoustic backscatter measured by the ADCPs to reveal modifications of the diurnal signal that is the result of DVM. In general, acoustic backscatter is explained by either suspended sediments (e.g., Wegner et al., 2005) or planktonic organisms (e.g., Petrusevich et al., 2020). Frazil ice crystals, which form during new ice growth within the coastal polynyas, also generate an enhanced acoustic backscatter (e.g., Dmitrenko et al., 2010a). However, because of DVM, the acoustic backscatter produced by zooplankton is more complex and easily recognizable from the backscatter generated by sediment particles and frazil ice crystals (Stanton et al., 1994). Moreover, ADCPs, unlike echo sounders, are limited in deriving accurate quantitative estimates of zooplankton biomass (Lemon et al., 2001, 2008; Vestheim et al., 2014). This is mainly due to calibration issues (e.g., Lorke et al., 2004) and to the beam geometry (Vestheim et al., 2014). Instead of absolute abundance values, we, therefore, work with the mean volume backscatter strength (MVBS) in decibels (dB) from the acoustic backscatter echo intensity following the procedure described by Deines (1999) and





updated by Mullison (2017). In what follows, we used ADCP acoustic backscatter from below the surface water layer where

the MVBS appears to be contaminated by the reflection of the acoustic signal from the air-water and/or ice-water interface.

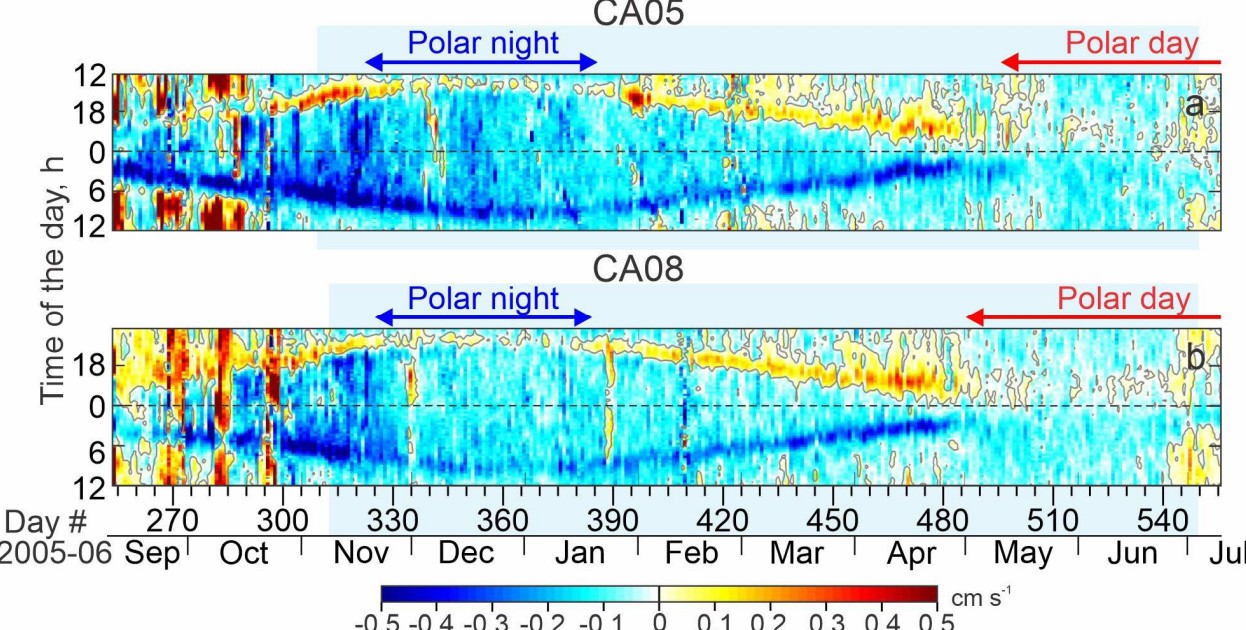

**Figure 8**: Actograms of ADCP-measured vertical velocity (cm s$^{-1}$) for (**a**) CA05 and (**b**) CA08 averaged for 39 to 63 m depth. Positive–negative values correspond to the upward–downward flow. Blue shading highlights sea-ice concentrations exceeding 90%. Red and blue arrows at the top indicate the periods of polar day and polar night, respectively.

To quantify the total sky illumination at each site we used the skylight.m function from the astronomy MATLAB package (Ofek, 2014). The irradiance values were estimated at the ice-free surface or under the ice. Under-ice illumination was modeled using the exponential decay radiative transfer model (Grenfell and Maykut, 1977; Perovich, 1996). Transmittance through the sea ice and snow cover to depth z in the ice was calculated using the following equation: $T(z) = i_0 e^{-K_t Z}$, where i0 is the fraction of the wavelength integrated incident irradance transmitted through the top 0.1 m of the surface layer, and $K_t$

is the total extinction coefficient in the snow or sea-ice cover. The values adopted for the sea-ice and snow cover were $i_0 = 0.63$ and $K_t = 1.5$ as well as $i_0 = 50.9$ and $K_t = 0.1$, respectively (Grenfell and Maykut, 1977). For computing under-ice irradiance in the Laptev Sea (Figures 4b and 5b), we used ADCP-derived estimates of sea-ice thickness from Belter et al. (2020). In the Beaufort Sea, we used PIOMAS-derived thickness. In both cases, snow depth came from AMSR-E. We only accounted for the sea ice and snow cover if the sea-ice concentration exceeded 90%. This implies that if sea-ice

concentration was less than 90%, the sea ice cover was not accounted for when light transmission was calculated. Due to a high uncertainty in cloud cover data retrieved from the Aqua/MODIS at high latitudes (Khanal and Wang, 2018), this model does not account for the variation in solar irradiance from clouds.



Time-series of MVBS (Figures 4–7c–g) and surface layer irradiance (Figures 4–7b), computed from the PIOMAS and ADCP estimates of sea-ice thickness and snow depth from AMSR-E, are presented in the form of actograms. Variations during a day-long period are presented along the vertical axis of the actogram, while the long-term patterns of diurnal behavior can be assessed following the horizontal axis (e.g., Leise et al., 2013; Last et al., 2016; Petrusevich et al., 2016, 2020; Hobbs et al., 2018; Dmitrenko et al., 2020, 2021). For the actograms of irradiance we introduced an artificial visual boundary on the irradiance color scheme at 1 lux (gray to orange), which is the threshold that corresponds to irradiance during the deep twilight. In addition, the vertical velocity actograms were calculated in the Beaufort Sea for the same depths as MVBS actograms, and were averaged through the subsurface water layer (Figure 8). Positive velocities are associated with the upward movement of particles.

In this study, we derive the total lead area (*LA*) from the MODIS lead product within a predefined reference circle (radii R of 5 km and 15 km) around a given mooring location (Anabar, Khatangar, CA05, CA08; Table 1). LA is given as the aerial sum of all pixels that were classified as a lead (hence, not including artefacts). The corresponding lead fraction (*LF*) is then the relative proportion of the daily *LA* with respect to the total area comprised by each circle (i.e., 78.54 km² for *r* = 5 km, and 706.86 km² for *r* = 15 km). Overall, both metrics serve as an indicator for the presence of thin ice and open water in proximity of the moorings during winter.

## 3 Results

Annual actograms of the modeled under ice irradiance and MVBS were computed for various depths at both the Laptev Sea (Figures 4 and 5) and the Beaufort Sea (Figures 6 and 7) based on total sky irradiance, sea ice concentration, sea ice thickness, and snow depth. These actograms reveal a rhythm of activity with the diurnal cycle seen in the vertical axis and the yearlong variability of the diurnal cycle observed along the horizontal axis. The strength of acoustic backscatter is presented by colored contours and provides qualitative information on the concentration of scatters in the water column.

### 3.1 Light intensity triggering DVM

The MVBS diurnal signal follows the seasonal variability of the sun irradiance with the onset of the descent and ascent following sunrise and sunset, respectively. Actograms in Figures 4–7 show that for both the Laptev and Beaufort Seas, there is a level of light intensity that triggers DVM during the twilight period. The DVM starts when the light intensity during twilight exceeds the light threshold. In general, DVM stops when light intensity is above or below the light threshold 24 hours a day (Figures 4−7). For the Laptev Sea, visual qualitative inspection of the irradiance and MVBS actograms reveals that the light intensity at the surface triggering DVM during the twilight period is ~1 lux (Figures 4 and 5) that is equivalent to about $7.91 \cdot 10^{-3}$ W m$^{-2}$ and $1.85 \cdot 10^{-2}$ µmol photons m$^{-2}$ s$^{-1}$. The light intensity of 1 lux corresponds to irradiance shortly after the end of civil twilight (Gaston et al., 2014). This result is consistent with that revealed for the Laptev Sea by Dmitrenko et al. (2021). For the Beaufort Sea, comparing actograms of MVBS and light shows the light intensity triggering





DVM of ~0.1 lux (Figures 6 and 7) that corresponds to the sun irradiance shortly before the end of the nautical twilight or to

355  the moonlight at full moon with a moon altitude $<\sim-30°$ on a clear night (Gaston et al., 2014). This light threshold resembles that obtained for the Beaufort Sea upper continental slope by Dmitrenko et al. (2020).

**Figure 9**: Scatter plots show MVBS anomaly (dB) relative to the 7-day running mean for the ADCP upper most level as function of under-ice irradiance (in a decimal logarithm scale) for (**a**) Anabar, (**b**) Khatanga, (**c**) CA05, and (**d**) CA08. Red

360  lines show linear regression. Blue numbers show estimates of under-ice irradiance (lux) triggering DVM. It is derived from the linear regression at the zero MVBS anomaly with confidence intervals computed for the 95% confidence level. $R$ is a linear correlation between MVBS anomaly and under-ice irradiance. Confidence intervals computed for the 95% confidence level.



More precise quantitative estimates of the light threshold triggering DVM are conducted based on the linear regression between simulated light irradiance as the independent variable (Figures 4–7b) and MVBS as the dependent variable (Figures 4–7c). The dependant variable (MVBS from the ADCP upper most level) was preliminary filtered with a 7-day running mean to retain the true signal representing DVM. We suggest that the light threshold triggering DVM corresponds to the zero anomaly of the filtered MVBS, which is derived from the linear regression (Figure 9). All statistical estimates are provided for the 95% confidence level.

Our analysis shows statistically significant negative correlations between light and MVBS; however, the correlation coefficients for the Laptev Sea (−0.15±0.01 and −0.22±0.01 for Anabar and Khatanga, respectively) are significantly lower than those for the Beaufort Sea (−0.29±0.02 and −0.44±0.01 for CA05 and CA08, respectively). Note that all the correlations are statistically significant. This difference is explained by two factors, as we discuss in detail in sections 3.2 and 3.4: (i) there is no DVM in the Laptev Sea during polar night, and (ii) compared to the Beaufort Sea, DVM in the Laptev Sea is impacted by polynya to a greater extent.

The linear regression analysis confirms different light thresholds triggering DVM in the Laptev and Beaufort Seas (Figure 9). In the Laptev Sea, the linear regression reveals the light threshold of $10^{0.52±0.35}$ lux and $10^{−0.10±0.21}$ lux that correspond to 3.3 lux and 0.8 lux for Anabar and Khatanga, respectively (Figures 9a and 9b). Taking into account the confidence intervals, the difference between Anabar and Khatanga only slightly exceeds the confidence level because of a high scatter in Figures 9a and 9b. For the Beaufort Sea, we estimate the light threshold to $10^{−0.53±0.13}$ lux and $10^{−0.40±0.08}$ lux that correspond to 0.3 lux and 0.4 lux for CA05 and CA08, respectively (Figures 9c and 9d). The difference between CA05 and CA08 is statistically insignificant. Overall, our results show that the light threshold at the sea surface or below the ice triggering DVM in the Laptev Sea is about two to ten times higher compared to the Beaufort Sea.

## 3.2 Diurnal Signal of the Mean Volume Backscatter Strength During Polar Night

During polar night (i.e. when the sun is below the horizon 24 h a day), the DVM stops when the light intensity does not exceed the light threshold through the entire 24 h. This is justified based on the disappearance of the difference in MVBS between midnight and noon. For the Laptev Sea, MVBS data shows that during a majority of polar night, there is no difference between acoustic scattering at midnight and noon as soon as under-ice light intensity falls below the light threshold revealed in section 3.1 (Figures 9a and 9b). This indicates vanishing of DVM observed through the water column resolved with ADCP data at both Anabar and Khatanga (Figures 4 and 5). For example, in the Laptev Sea during ±7 days to winter solstice, through the subsurface water layer there is no correlation between MVBS anomaly and light intensity, which does not exceed $10^{−2}$ lux (Figures 10a and 10b). This light intensity is far below the light threshold triggering DVM in the Laptev Sea. In the Beaufort Sea, when under-ice irradiance falls below 0.2 lux during two weeks around winter solstice,



MVBS maintains difference between midnight and noon reducing around noon by ~10 to 20 dB compared to midnight (Figures 10c and 10d). Moreover, correlation between the MVBS anomaly and light intensity through the subsurface water layer is estimated to be −0.68 and −0.74 for CA05 and CA08, respectively (Figures 10c and 10d). This indicates the occurrence of DVM during polar night while the sun is below the horizon 24 h a day (Figures 6 and 7). However, the DVM signal is traceable during the entire polar night only at the upper level down to 31−32 m depth (Figures 6c, 6d, 7c, and 7d).

For the deeper water layer, the DVM signal disappeared after winter solstice (Figures 6e−g and Figures 7e−g).

For the Beaufort Sea, another set of actograms was generated for the ADCP-measured vertical velocity averaged for 39 to 63 m depth (Figure 8). During polar night, vertical velocity actograms show a nearly symmetrical diurnal pattern around astronomic midnight that is consistent with the backscatter actograms. Net upward flow was observed daily before the astronomic midnight. Note that in the Laptev Sea, actograms for the ADCP-measured vertical velocity did not reveal DVM.

In summary, in the Laptev Sea there is no DVM to observe during the major portion of polar night. In the Beaufort Sea, DVM is conducted throughout the entire polar night. In terms of the light conditions during polar night, the study area in the Beaufort Sea is substantially different from that in the Laptev Sea. In the Beaufort Sea (mooring array at ~71°N, Table 1), civil twilight (the Sun's disk at most 6° below the horizon) is observed through the entire polar night after sunset and before sunrise. Duration of civil twilight ranges from ±3 h at noon on 22 November 2005 (beginning of polar night at 71°N) to

±1.33 h on 22 December 2005 (winter solstice), and to ±2.65 h on 22 January 2006 (end of polar night at ~71°N). This means that during polar night, civil twilight occurs at CA05 and CA08, illuminating the subsurface under-ice water layer even during winter solstice from ~0.2 lux at noon to 0.05 lux at the end of civil twilight (Figures 6b, 7b, 10c, and 10d). In contrast, in the Laptev Sea at ~74.5°N (Table 1), there is no civil twilight during the portion of polar night from 30 November 2007 to 13 January 2008 when the sun remains >6 degrees below the horizon. During this time, the light intensity

below the ice does not exceed $10^{-2}$ lux observed during full moon events (Figures 6b, 7b, 10a, and 10b). However, the difference in DVM between the Laptev and Beaufort Seas during polar night is not entirely assigned to the different latitudinal position, but also to the different magnitude of light threshold triggering DVM (see section 3.1).





**Figure 10:** Time series of MVBS anomaly (black, dB) relative to the 7-day running mean for the ADCP upper most level and under-ice irradiance (lux) in a decimal logarithm scale (red) for 7 days before and after winter (left) and summer (right) solstices for (**a**, **e**) Anabar, (**b**, **f**) Khatanga, (**c**, **g**) CA05, and (**d**, **h**) CA08. Correlation $R$ between the MVBS anomaly and under-ice irradiance estimated for the 95% confidence level. Asterisks highlight statistically insignificant correlations.





### 3.3 Diurnal Signal of the Mean Volume Backscatter Strength During Polar Day

The response of MVBS diurnal pattern to enhanced irradiance during polar day (the sun is above the horizon 24 h a day) in the Laptev Sea differs from that in the Beaufort Sea. During polar day, the MVBS diurnal rhythm in the Laptev Sea completely vanished (Figures 4, 5, 10e and 10f). This is attributed to enhanced sunlight with an increase in the midnight under-ice irradiance to >15 lux from the beginning of May (Figures 4b and 5b) to ~$10^{4.7}$ lux at summer solstice with diurnal variability of ~ ±5 lux (Figures 10e and 10f). This level of light intensity is far above the threshold triggering DVM. During

±7 days to summer solstice, there is no statistically significant correlation between MVBS and light intensity for the subsurface water layer at both Anabar and Khatanga (Figures 10e and 10f) indicating no DVM to occurs. Actograms in Figures 4c−g and 5c−g do not show DVM during polar day as well. Note, however, that the DVM signal in the Laptev Sea has been already ceased advancing polar day (Figures 4 and 5). A surface-intensified enhancement of MVBS at Anabar during the first part of June 2008 (Figures 4c−g) seemed to be impacted by intrusions of turbid water generated by the Lena

River runoff.

In the Beaufort Sea, light intensity below the ice is rather similar to that of the Laptev Sea (Figures 4−7b). CA05 and CA08 became ice-free shortly after summer solstice lagging ice retreat in the Laptev Sea by about one-and-a-half month (Figures 4−7a). Sea ice retreat at CA05 and CA08 results in a 30-fold increase of the surface layer irradiance up to the level simulated for the Laptev Sea (Figures 6b and 7b). Around summer solstice at CA05, statistically significant correlation between MVBS

and light intensity ($R = −0.34$) indicates occurrence of DVM at the subsurface water layer (Figure 10g) while it is not obvious from the actogram in Figure 6c. In the deeper water layer, DVM below the ice is confirmed until summer solstice by the difference in MVBS from the midnight to noon (Figures 6d−g). However, this signal is gradually vanished to the beginning of July, when CA05 became ice-free (Figures 6a). For the subsurface layer at CA08, there is no correlation between MVBS and light intensity around summer solstice (Figure 10h). However, MVBS actograms for the deeper layer

clearly show DVM during polar day until the ice started to retreat before the beginning of July 2006 (Figures 7d−g).

Actograms for the ADCP-measured vertical velocity in the Beaufort Sea show that the DVM pattern is hardly recognizable after the beginning of May 2006 when the polar day began (Figure 8). However, net downward and upward flow remains traceable before and after mid-night, respectively, until the end of June 2006 (Figure 8). Note that the range of vertical velocity ~ ±0.1 cm s$^{-1}$ associated with this flow may be within the noise level of velocity measurements.

### 3.4 Diurnal Signal of the Mean Volume Backscatter Strength During Transitional Periods

In the Laptev Sea, during the fall transition from polar day to polar night (approximately to the beginning of November 2007), the occurrence of the MVBS diurnal pattern corresponds to a decrease in the surface layer irradiance after sunset below 3.3 lux and 0.8 lux for Anabar and Khatanga, respectively (Figures 9a and 9b). This irradiance threshold is associated with light intensity during the deep twilight (see section 3.1 for more details). The MVBS diurnal signal roughly follows the





irradiance threshold with acoustic backscatter enhanced during the dark. Negative correlation between MVBS and irradiance during ±7 days to autumnal equinox at both Anabar and Khatanga ($R = -0.63$; Figures 11a and 11b) confirms DVM during transition from polar day to polar night. After the beginning of polar night, once the under-ice irradiance falls below irradiance threshold for 24 h a day, the MVBS diurnal pattern vanishes lagging behind the beginning of polar night by about 2 weeks (Figures 4b–g and 5b–g). This is a period when the twilight still persists after sunset. In the Laptev Sea, DVM during fall (and spring) was well persistent down to 24 m at Anabar (Figures 4c−f) and to 30 m depth at Khatanga (Figures 5c−f).

In the Beaufort Sea, transitional period from polar day to polar night in September to November 2005 is characterized by MVBS diurnal patterns similar to those in the Laptev Sea (Figures 6 and 7). However, the light threshold triggering DVM is about two times to one order of magnitude lower compared to the Laptev Sea (see section 3.1). In the Beaufort Sea, seasonal DVM during transitions from polar day to polar night (and vice versa) was observed through the entire depth range of the ADCP profiling (Figures 6c−g and 7c−g). During ±7 days to autumnal equinox, negative correlation between MVBS and irradiance clearly indicates DVM at CA08 ($R = -0.45$; Figure 11d). In contrast, positive correlation at CA05 ($R = 0.37$; Figure 11c) suggests the reversing DVM through the sub-surface water layer as also follows from actogram in Figure 6c. Moonlight and/or upwelling seem to cause the DVM reversing at CA05 during this time. In contrast to the Laptev Sea, DVM in the Beaufort Sea was not disrupted following the beginning of polar night as discussed in section 3.3 (Figures 6, 7, 10c, and 10d).

The MVBS diurnal pattern resumed (Laptev Sea) and enhanced (Beaufort Sea) during spring transition from polar night to polar day. This corresponds to a daily increase in the under-ice irradiance to the level exceeding light threshold revealed in section 3.1. Negative correlation between MVBS and irradiance during ±7 days to spring equinox for all four moorings (Figures 11e−h) indicates DVM in the subsurface water layer. However, the relationship between irradiance and MVBS was disrupted approaching and following the beginning of polar day in the Laptev and Beaufort Seas, respectively.

In the Laptev Sea, the DVM signal shallowed vanishing at 24−27 m and 35 m depth at Anabar and Khatanga, respectively (Figures 4f, 4g, and 5g). For the overlying water layer, a daily cycle in MVBS was mimicked by the diurnal cycle in the sun irradiance until March 2007 (Figures 4 and 5). Onward, the MVBS diurnal signal ceased lagging polar day by about one month to one-and a half month for Khatanga and Anabar, respectively (Figures 4 and 5). At Anabar, DVM disappeared in mid-March preceding full moon event #7 (Figure 4). At Khatanga, DVM abruptly vanished on 26 March 2008, approximately 1-month prior to the start of polar day when the MVBS diurnal signal would be expected to end. In fact, at Anabar and Khatanga, there was no difference in MVBS between midnight and noon indicating disappearance of the diurnal rhythm already before the beginning of polar day. Note that from the mid-March 2007 the polynya started to develop in the Laptev Sea (Figure 2), but both moorings were still located below the pack ice of ~1.7 m thick (Figures 4a and 5a).





**Figure 11:** Time series of MVBS anomaly (black, dB) relative to the 7-day running mean for the ADCP upper most level and under-ice irradiance (lux) in a decimal logarithm scale (red) for 7 days before and after autumnal (left) and spring (right) equinox for (**a**, **e**) Anabar, (**b**, **f**) Khatanga, (**c**, **g**) CA05, and (**d**, **h**) CA08. Correlation $R$ between the MVBS anomaly and under-ice irradiance estimated for the 95% confidence level.



In the Beaufort Sea, the MVBS diurnal signal extended relative to the beginning of polar day by about one-and-a-half month. At 15−16 m depth, diurnal signal followed the 0.3−0.4 lux threshold and vanished once the under-ice irradiance exceeded light threshold throughout the day shortly before the start of polar day at the beginning of May 2006 (Figures 6c and 7c). During this time, CA05 and CA08 were still located beneath the pack ice that was ~2 m thick (Figures 6a and 7a). In contrast to the subsurface water layer, MVBS diurnal pattern at deeper water layers maintained identity during the first half of polar day and vanished shortly after summer solstice (Figures 6d−g and 7d−g). Here DVM was disrupted only at the end of June 2007 once the sea ice started to deteriorate after 27 June 2006 (Figures 3, 6, and 7).

## 4. Discussion

### 4.1 On the light threshold triggering DVM in the Laptev and Beaufort Seas

Our acoustic backscatter analysis shows that the light threshold triggering DVM in the Laptev Sea is about two to ten times higher compared to the Beaufort Sea. However, this analysis is limited by lack of zooplankton observations. It is logistically difficult to sample zooplankton in the seasonally ice covered and remote areas of the Arctic shelf, hence it is unclear which zooplankton species were likely to undergo DVM. Acknowledging this deficiency, the difference of zooplankton response to irradiance in the Laptev and Beaufort Seas could potentially be attributed to three different factors described below.

First, different zooplankton species respond to specific wavelengths (Cohen et al., 2015; Cohen and Forward, 2002). It is possible that different zooplankton communities occurred in the Laptev and Beaufort Seas, reacting differently to wavelengths at a specific time of the year. It is also possible that lower turbidity levels and CDOM concentrations in the Beaufort Sea resulted in different wavelengths in the Beaufort and Laptev Seas.

Second, although some studies have not found any impact on the response of zooplankton to light in relation to temperature (e.g., Cohen and Frank, 2006), others have found that krill have higher light sensitivity at cold temperatures (J. Cohen, University of Delaware, personal communication). Moreover, temperature can also induce changes in the irradiance sensitivity of some zooplankton, and a temperature-induced change in the animal's phototactic reactions may impact the DVM response to irradiance (Bandara et al., 2021). The lower surface temperatures prevailing in the Beaufort Sea (Figures S3 and S5) could thus partly explain the difference in seasonal DVM patterns.

The third and most likely reason behind the different DVM response to irradiance is related to bottom depth (<50 m in the Laptev Sea and 200−400 m in the Beaufort Sea). In the Arctic Ocean, including its marginal seas, DVM are documented for several large-sized abundant Arctic copepods - *Metridia longa*, *Calanus glacialis*, smaller sized *Pseudocalanus* spp., mysids, and a chaethognath *Parasagitta elegans* (Runge and Ingram, 1991; Daase et al, 2008; Vestheim et al., 2013). Most of the listed species are inhabitants of relatively deep Arctic shelf regions with depths > 50 m (Conover and Huntley, 1991; Kosobokova et al., 1998; Darnis et al., 2022; Skjoldal and Aarflot, 2023). In the shallower regions, like the sites where



Anabar and Khatanga were deployed in the Laptev Sea (30-40 m), only copepods *Pseudocalanus* spp. and mysids could be regarded as migrators, but they are numerous in these regions only during the ice-free summer period (*Abramova*, 1999). In the deeper Beaufort Sea, in contrast, all the listed species-migrators are abundant during the entire year and could potentially perform DVM depending on irradiance cycle. Due to the pronounced differences in the composition of zooplankton

communities in the shallow Laptev Sea and much deeper region of the Amundsen Gulf in the Beaufort Sea, one can expect quite different patterns of DVM in these two regions that we, in fact, do observe using acoustic methods. Moreover, the very shallow Laptev Sea restrained the amplitude of the DVM, and it is possible that at low light conditions, the difference in irradiance between the surface and seafloor was not strong enough to trigger DVM. It is also possible that planktivorous fish such as polar cod migrate to deeper areas in winter, a period during which they prefer warmer and deeper Atlantic waters

(Geoffroy et al., 2011), thus avoiding the location of the Laptev Sea moorings. The absence of visual predators during that period would thus hinder the need for DVM during the polar night in the Laptev Sea because DVM are a combination of grazing needs and predator avoidance.

### 4.2 Sea-ice impact on DVM

Actograms in Figures 4–8 show that DVM in the Laptev and Beaufort seas was in general controlled by light attenuated by

the first-year pack ice. This pattern, however, was deviated about 1.5 month before and after the beginning of polar day in the Laptev and Beaufort Seas, respectively. We speculate that the disappearance of DVM in the Laptev Sea before the beginning of polar day is attributed to the polynya opening. In the Beaufort Sea, the DVM has been extended over the polar day to summer solstice. We attribute this extension to light attenuation by the first-year pack ice. In this section, we examine these suggestions in more detail using sea-ice satellite imagery (Figures 12 and 13). Moreover, for elucidating the impact of

sea-ice disruptions on DVM in the Laptev and Beaufort seas, we use time series of correlations between subsurface MVBS and irradiance.

#### 4.2.1 Polynyas and leads

By the end of polar night and until mid-May 2007, in the Laptev Sea Anabar was located below the first-year pack ice of ~1.7 m thick. At Khatanga, the first-year pack ice started deteriorating before the end of June 2008 (Figure 2). In mid-March,

the southeasterly winds up to 8 m s$^{-1}$ forced the development of the coastal polynya in ~15 km to Anabar (Figures 2 and S2b). This corresponds to disappearance of DVM at Anabar (Figures 4c−g). Onward on 26 March and on 28−29 March 2008, two consecutive events of southwesterly winds generated polynya open water extended ~30 km off the landfast ice edge and ~15−30 km to Anabar and Khatanga where the ice cover was strongly deteriorated (Figures 2 and 12). This is consistent with abrupt cessation of DVM at Khatanga (Figures 5c−g) that was accompanied by the bottom-intensified

increase of MVBS by ~45 dB. Strong southeasterly winds during the second part of April 2008 (Figure S2b) extended polynya area in the southeastern Laptev Sea, and to the end of April, Anabar was located below the polynya open water



(Figures 2 and 4a). Thus, from the analysis of wind and satellite imagery we reveal that the DVM disruptions in the Laptev Sea are likely attributed to the polynya openings.

**Figure 12**: The MODIS true-color satellite imagery for 27 March to 5 April 2008 shows the evolution of polynya in a relatively cloud-free area northward of the Lena Delta (Laptev Sea). Red crosses mark moorings Khatanga and Anabar. Blue crosses depict the spatial node where 2-m air temperature and 10-m wind were derived from ERA5 (Figures S2a and S2b, respectively). Red arrows show daily mean wind for the day the image was taken. No arrows indicates no wind.




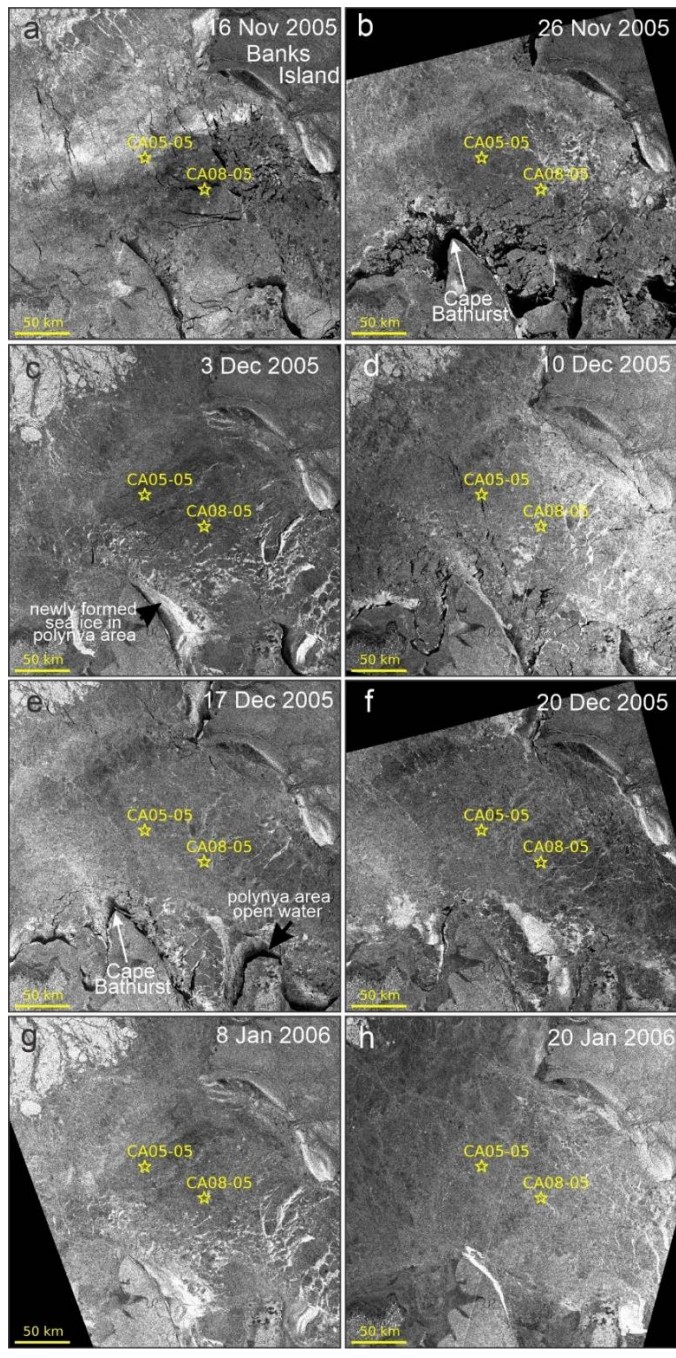


**Figure 13**: RADARSAT-1 ScanSAR satellite images show the evolution of ice conditions in Amundsen Gulf in (**a, b**) November and (**c–f**) December 2005, and (**g, h**) January 2006. Yellow stars depict moorings CA05 and CA08. Low radar backscatter area corresponds to calm open water in leads and polynyas. The gray strips are associated with a thin sea-ice newly formed in leads and polynyas.





**Figure 14**: Correlation *R* between the mean volume backscatter strength at the ADCP upper most level and irradiance computed for the 1-day moving window (red) with their 7-day running mean (black) for (**a**) CA05, (**d**) CA08, (**g**) Anabar, and (**j**) Khatanga. Gray shading highlights statistically insignificant correlations at the 99% confidence level. Blue arrows at the bottom indicate the period of polar night. Blue shading highlights disruptions of DVM attributed to polynyas and leads (**b, c, e, f, h, i, k, and l**). Time series of the lead area (blue, km$^2$) and fraction (red, %) derived from the MODIS ArcLead data set (Reiser et al., *2020*) for the 5-km (**b, h, e, and k**) and 20-km (**c, i, f, and l**) radius circles centered at mooring positions.



We suggest that the deviations or disruptions of correlation between subsurface MVBS and light can be the measure of polynya impact on DVM. In the Laptev Sea, during a major portion of polar night when there is no twilight to observe, there
is no statistically significant correlation between MVBS and light (Figures 14g and 14j) indicating no DVM. Transition from polar night to polar day is characterized by statistically significant correlation < −0.35 revealing DVM. At Anabar, this relationship was disrupted in mid-March following several consecutive events of polynya opening nearby Anabar (Figures 2, 12, 14h, and 14i). Further onward from the beginning of April 2008, correlation is decreasing consistently with increasing of the lead fraction (i.e., open water and thin ice) derived from MODIS thermal infrared imagery (Figures 14h and 14i). Finally,
correlation falls to zero on ~26 April (Figure 14g) when Anabar became ice-free (Figure 2). At Khatanga, during the last decade of March 2008, consecutive events of polynya opening 5−20 km to the station (Figures 2, 12, 14k, and 14l) completely disrupted correlation between MVBS and light to the beginning of April (Figure 14j). This is evidenced for the polynya impact on DVM, while Khatanga became ice-free only in mid-July 2008 (Figure 2). Similarly to these results, Dmitrenko et al. (2021) reported that the formation of the Laptev Sea coastal polynya in late March 1999 caused DVM to
abruptly cease near the landfast ice edge, while DVM persisted through spring to the start of polar day at the onshore mooring.

In the Beaufort Sea, during the major portion of polar night, negative correlations between MVBS and irradiance were statistically significant (Figures 14a and 14d) indicating DVM as discussed in section 3.2. However, there were several short-term events when negative correlations become statistically insignificant. At CA05, disruption of DVM in December
2005 (Figure 14a) is attributed to deterioration of ice pack and formation of leads at the vicinity of CA05 (Figures 13c−f, 14b, and 14c) forced by north-easterly winds (Figure S2d). At CA08, two lead events during the second part of November 2005 and in January 2006 (Figure 13a, 13b, 13g, 13h, 14e, and 14f) deviated correlation between MVBS and irradiance disputing DVM (Figure 14d). The polynya opening in 17−20 March 2006 at the vicinity of CA05 (Figures 3, 14b and 14c) did not disrupt DVM at 16 m depth. However, the deeper water layer shows deviations in DVM attributed to polynya
opening (Figures 6d–g). Note, however, that lead events from the end of March to mid-April did not deviate DVM at CA05 and CA08. During these events DVM was totally driven by light.

Our observations suggest that DVM is not solely controlled by daily changes in irradiance, as there were no abrupt changes to the under-ice light conditions while the Laptev Sea polynya was open during March–April 2008, and both Anabar and Khatanga were located at a distance to polynya open water (Figures 2, 3a, 4a, and 12). In the Beaufort Sea, appearance of
open water is mainly attributed to deterioration of ice pack and formation of leads at the vicinity of CA05 and CA08 (Figures 3 and 13). Only the open water event in 17-20 March 2006 can be attributed to polynya opening, but CA05 was located at a distance to polynya open water (Figure 3).

Following Dmitrenko et al. (2021), we suggest that formation of polynyas and leads generates far-field effects on zooplankton behavior. Since the opening of the polynya and leads did not cause any immediate change in irradiance at





moorings, the observed deviations of DVM could potentially be explained by predator-avoidance behavior of zooplankton. Polar cod (*Boreogadus saida*) is one of the main predators of the zooplankton taxa likely responsible for the observed DVM (e.g., Buckley and Whitehouse, 2017; Cusa et al., 2019; Bouchard and Fortier, 2020; Aune et al., 2021; Geoffroy et al., 2023). Among other habitats, polar cod is known to occupy surface waters over shallow coastal continental shelves in winter and spring (e.g., Gradinger and Bluhm, 2004; David et al., 2016). Polynya and lead openings may increase zooplankton

mortality by increasing the predation pressure from polar cod because more light from polynya openings increases the prey detection distance of polar cod, improving its capacity to forage on zooplankton (Bouchard and Fortier, 2008; Jönsson et al., 2014; Varpe et al., 2015; Langbehn and Varpe, 2017; Cusa et al., 2019). The zooplankton possibly stopped conducting DVM and remained closer to the seafloor, as a predator-avoidance behavioral response to the increase in polar cod abundance in the photic zone near the surface.

In the Laptev Sea, DVM disruptions were observed in response to polynya openings during the entire transitional period from polar night to polar day (Figures 2, 4, 5, and 14g–l). In the Beaufort Sea, DVM was partially deviated in response to formation of leads in November to January. However, in contrast to the Laptev Sea, DVM in the Beaufort Sea was not deviated in response to several significant events of open water observed in March−April 2006 (Figures 3 and 14a−f). Despite the open water at the vicinity of moorings, negative correlations between MVBS and irradiance remain statistically

significant (e.g., Figures 11g and 11h). We attribute this difference to the seasonal behavior of polar cod. Aggregations of polar cod in the Amundsen Gulf were observed from December to mid-March (Geoffroy et al., 2011). Dissipation of the polar cod aggregation to March−April eliminates the predators' risk, and zooplankton resumes DVM during polynya and lead events as follows from Figures17a−f. This suggests that a low predation pressure environment in the Amundsen Gulf during spring favors zooplankton to conduct DVM during formation of polynyas and leads. Another possible reason is

seasonal migration of polar cod to deeper and warmer Atlantic waters (Geoffroy et al., 2011) located in the Amundsen Gulf below 220–250 m depth (Figures S3d, S5a, and S5d).

### 4.2.2 First-year pack ice

Our data from the Beaufort Sea show that, during polar day 2006, DVM ceased only at 15 m and 16 m of depth at CA08 and CA05, respectively (Figures 5c and 6c). In the underlying water layer, DVM continued until the beginning of July 2006

(Figures 5d–g and 6d–g). It seems that zooplankton were conducting DVM, but they were still avoiding relatively well-illuminated subsurface water. This is in line with predator-avoidance behavior during transitional seasons, but without seasonal modulation, because the sun is above the horizon 24 h a day. We speculate that the light intensity at depths exceeding 15–16 m was below the threshold of predator perception, allowing DVM at these depths to extend over polar day to summer solstice 2006.

We suggest that the polar day DVM in 2006 was likely generated by attenuation of light caused by the first-year pack ice of ~1.8 m thick. During the first part of polar day to summer solstice, the subsurface under-ice water layer in the Beaufort Sea



was illuminated to $10^3$ lux that is ~100 times lower than in the Laptev Sea (Figures 10e–h). This difference is entirely explained by light attenuation from the ice pack that was persisted at CA05 and CA08 until the very beginning of July, and retreated onward shortly after summer solstice. Thus, our results from the Beaufort Sea highlight an important role of sea ice

in promoting DVM during polar day when sunlight illuminates the surface water layer 24 h a day. Note that for the Beaufort Sea continental slope, *Dmitrenko et al.* (2020) reported DVM prolongation toward polar day due to the sea ice shading the under-ice water layer by the multiyear Greenland pack ice.

**4.3 DVM disruptions related to wind-forced water dynamics and moonlight**

Our results revealed that water dynamics temporally impact DVM by disrupting the diurnal rhythm. In the Laptev Sea, two

wind-driven storms in the end of September and beginning of October 2007 with northerly winds exceeding 20 m s$^{-1}$ resulted in vertical mixing of the water column down to the seafloor at Anabar (Hölemann et al., 2011). These storm events completely modified initial vertical distribution of temperature and salinity shown in Figures S3a generating positive downward heat flux and negative salt flux (Hölemann et al., 2011). The actograms of MVBS at Anabar show that following these two storms, DVM was disrupted down to the seafloor (Figures 3c–g). At Khatanga, MVBS signal was becoming

noisier, but DVM remained traceable (Figures 4c–g).

In the Beaufort Sea, three consecutive upwelling-favorable storms with north-easterly winds exceeding 10 m s$^{-1}$ were observed from the end of September to the end of October 2007 before freeze-up onset (Figure S2d). Upwelling of nutrient-rich, Pacific-origin water to the surface at Cape Bathurst has been commonly observed (Williams and Carmack, 2008). For example, upwelling of cold and saline water on the western flank of the Amundsen Gulf was observed in October 2006

(Figures S5d and S5e), about 2.5 month after CA05 and CA08 were recovered in July 2006. In winter, the upwelling-favorable wind forces the mobile pack ice to move off the stationary landfast ice edge forming the Cape Bathurst polynya.

The actograms of vertical velocity from CA08 show deviations of DVM consistent with three upwelling-favorable storms in September–October 2005 (Figures 8b and S2d). A net upward flux typical for coastal upwelling seems to be superimposed on DVM partially masking the DVM signal. At CA05, the deviations imposed by coastal upwelling were less recognisable

(Figure 8a). Actograms of MVBS for CA05 and CA08, however, do not show DVM disruptions that can be attributed to the upwelling-favorable storms.

Wind forcing also plays a role in formation of enhanced acoustic scattering on the top of MVBS signal generated by DVM. In the Laptev Sea, we observed high MVBS events that can be attributed to the frazil ice crystals and riverine suspended sediments (e.g., Wegner et al., 2005; Dmitrenko et al., 2010a; Ito et al., 2020). An enhanced acoustic backscatter was

recorded following the wind-driven open water events at Anabar during the very end of April and mid-May 2008 when sea ice concentration fell below 90% (Figures 4a, 4c, and S2a). The frazil ice was likely generated by new ice growth within the polynya open water during these two periods when air temperatures remained relatively cold (–12°C and –7°C in April and May 2008, respectively; Figure S2a).





In contrast to MVBS events in April–May, the enhanced acoustic scattering in July–August at Anabar and Khatanga (Figures 4c and 5c) seems to be attributed to intrusions of turbid water associated with river runoff. The Lena River plume is enriched with suspended sediments (e.g., Pivovarov et al., 1999; Wegner et al., 2005). We suggest that intrusions of turbid riverine water generated high MVBS at Anabar and Khatanga in July–August 2008. During July 2008, northeasterly winds (Figure S1e) resulted in divergence of the Lena River plume to the southeastern Laptev Sea, the area where Anabar and Khatanga were located (Figure S3a, S3b, and S4b; for more details see Dmitrenko et al., 2010), generating enhanced MVBS at the front of the Lena Delta. For the southeastern Beaufort Sea, the southwesterly winds dominated during summer 2006 (Figures S1b and S1c) are suggested to force eastward transport of the Mackenzie River plum off the Amundsen Gulf where CA05 and CA08 were deployed. Hence, the signature of riverine water suspended sediments is not traceable at MVBS actograms during summer 2006 (Figures 6 and 7).

Finally, MVBS actograms in Figures 4–7 suggest no deviation of the MVBS diurnal pattern that can be associated with moonlight. During full moons, the midnight under-ice irradiance increased up to 0.01 lux in October and to 0.001 lux in February–March (Figures 4–7a). This level of light intensity is far below the threshold revealed in section 5.1. Furthermore, the uncertain data on the total cloud cover from the Aqua/MODIS revealed high cloudiness until November when the sea ice thickness did not exceed 50 cm (Figures 4–7a) that makes any suggestions about the impact of moonlight on DVM before the beginning of polar night to be very speculative. Afterwards, during polar night, the mean cloudiness decreases to 5-10%, but gradually growing sea ice thickness and snow depth both increase light attenuation limiting under-ice light intensity to 0.001 lux. However, the moonlight impact on DVM cannot be completely discriminated. The DVM reversal at CA05 in the end of September 2005 was interpreted as a result of upwelling due to favorable wind forcing (Figures 6c and S2d). Alternatively, it can be explained by the moonlight during the full moon event #1 when the mid-night light intensity was slightly below 0.1 lux (Figure 11c). Note that during this period the total cloud cover varied from 20% to 100% (Figure 6b).

## 5. Conclusions and final remarks

Based on yearlong mooring observations from the southeastern shelves of the Laptev and Beaufort Seas, both of which are in the vicinity of the circumpolar Arctic coastal polynya, we conclude that DVM in both areas is influenced by the formation of coastal polynyas and leads, and the associated increase in light transmittance, though the light threshold triggering DVM in the Laptev Sea is about two times to one order of magnitude higher than what is observed in the Beaufort Sea. Along with a deeper and more southerly location of the mooring array in the Beaufort Sea, a lower light threshold allows DVM to occur in the Beaufort Sea during polar night whereas DVM ceases in the Laptev Sea. We attribute this different pattern in DVM to different zooplankton communities, different wavelengths and water temperatures, and/or a reduced predation pressure due to the Laptev Sea shallower waters.



In the Laptev Sea, during the spring transition from polar night to polar day, DVM ceased ~1 month prior to its expected cessation at the start of polar day. We associate this disruption with a predator-avoidance behavior of zooplankton conditioned by higher polar cod abundance attracted by the polynya opening. In the Beaufort Sea, leads cause DVM to deviate only during winter when aggregations of polar cod are observed. During spring, a low predation pressure environment favors zooplankton to conduct DVM during polynya and lead events. In the Laptev Sea, the acoustic backscatter during polar day, when DVM had stopped, is caused by scattering from suspended sediments within the Lena

River plume. In the Beaufort Sea, the first-year pack ice prolonged the occurrence of DVM toward the polar day due to the sea ice shading the under-ice water layer.

Overall, our results highlight the role of sea ice in disrupting synchronized DVM at the vicinity of polynyas and leads, but also in promoting DVM below the ice during polar day. In general, increasing light penetration through polynyas and leads reduces the under-ice habitat of zooplankton. In contrast, an extension of the ice-covered period favours the under-ice habitat

of zooplankton. This is in in agreement with the conclusions of Flores et al. (2023), who suggested the deepening of zooplankton in response to sea-ice decline. Given the fact that the size of the Laptev Sea and Cape Bathurst polynyas during winter tends to increase over the last two decades (Preußer et al., 2019), the cessation of DVM in the surrounding areas can be important for the ecology of Arctic zooplankton and may disrupt the biological pump, which has implications for carbon fluxes (e.g., Kelly et al., 2019; Archibald et al., 2019).

**Declaration of Interests**

The authors declare that they have no known competing financial interests or personal relationships that could have appeared to influence the work reported in this paper.

**Author Contribution**

ID and VP guided the overall research problem and developed methodology. ID, MG, and KK conceptualized this research.

ID, VP, AK, AP, and SK conducted formal analysis and data curation. SK performed field research. DBar allocated resources. ID, KK, AP, and MG wrote the original draft. ID, AP, KK, AK, MG, CB, and DBab revised and edited the original draft. ID, VP, and AK generated figures. ID and DBar supervised and administrated this project. DBar accomplished the funding acquisition. All authors contributed to the article and approved the submitted version.

**Data availability statement**

The ADCP data that support the findings of this study are openly available in Mendeley Data at https://doi.org/10.17632/75rk5kbnn4.1. They are in a binary instrumental format and can be assets using WinADCP, a



software package for use with RDI Acoustic Doppler Current Profilers for the interactive exploration, analysis and visualization of ADCP data. WinADCP documentation can be find at: https://www.comm-tec.com/Docs/Manuali/RDI/WinADCP%20User%20Guide.pdf.

**Funding**

Field research in the Laptev Sea was funded through the Bundesministerium für Bildung und Forschung projects "System Laptev Sea". Field research in the Beaufort was conducted as part of the Canadian ArcticNet field program. Funding for this work was provided by the Canada Excellence Research Chairs (CERC) program, Prof. Dr. Dorthe Dahl-Jensen. The work of KK was funded by the Russian Science Foundation (grant No. 23-17-00121) and carried out within the framework of the
state assignment of the IORAS (theme No. FMWE-2024-0021). AP was supported by the European Union's Horizon 2020 research and innovation programme under grant 101003472, as well as by the German Federal Ministry of Education and Research (Bundesministerium für Bildung und Forschung – BMBF) under Grant 03F0831C (CATS-Synthesis).

**Acknowledgements**

We dedicate this paper to our colleague, Dr. David Barber (1960–2022). Through his vision, leadership and endless efforts,
David was a visionary researcher with a passion for the Arctic, a scholar with an entrepreneurial spirit, and a generous mentor and our friend. We thank NASA and the Alaska Satellite Facility for granting access to RADARSAT-1 imagery used in this study.

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
