# Peer review of "Contrasting two major Arctic coastal polynyas: the role of sea ice in driving diel vertical migrations of zooplankton in the Laptev and Beaufort Seas"

_EGUsphere, 2024_

## Author Response (AR1)

In the following, the comments by Reviewer #1 and #2 are underlined and our responses to the comments are in normal characters. Modifications to the text are shown in quotation marks with bold characters indicating newly added text, and normal characters indicating text that was already present in the previous version.

**Reviewer #1**

We highly appreciate helpful comments and suggestions by Reviewer #1.

I have now read the manuscript, and I find it to be well written, clear, and with interesting and important new findings. The quality of the figures are excellent, and in general I find that the interpretation and conclusions drawn from the data are solid, but I do have a few comments / questions outlined below.

Thank you!

1. Line 33: «In the Laptev Sea, DVM does not occur during the polar night…". I would urge the authors to modify this statement, and rather say that based on their data and methodology, "DVM could not be detected". It is an important difference between the two. See also below

This statement in abstract was modified as follows: "*In the Laptev Sea,* **based on our data and methodology, DVM could not be detected**".

2. Line 38 (and several other places in the manuscript): based on the fact that the trivial name "polar cod" is used differently between the American and Eurasian side of the Arctic, I recommend to indicate in the beginning of the manuscript which species you refer to using also the latin name.

We added the latin name (*Boreogadus saida*) to the abstract and to introduction where the polar cod was mentioned for the first time. We also modified the first sentence of the third paragraph of section 2.1 as follows: "*In the Beaufort Sea, both zooplankton and polar cod* **(Boreogadus saida), also called Arctic cod in North America,** *conduct DVM from August until May…*".

3. Line 132: remove additional space between "Sea" and "has".

Implemented.

4. Section 3.2. Regarding DVM and polar night. The method used to reveal DVM is very coarse (but nevertheless appropriate), and unlikely to detect either small-scale DVM (see e.g. Ludvigsen et al 2018) or DVM by smaller organisms than detected by a 300kHz ADCP (by the way – can this instrument reliably detect fish? Will not the signal be too high due to the swim bladder, and hence be treated as an anomaly by the instrument?). I would argue that the authors include this in their discussion, and that they try to be more careful in their statements regarding when/if DVM is occurring. I agree with the authors that the data at hand does not indicate DVM, but that is very different from stating that there is no DVM. Conclusions are based on observations, and observations have limitations. And as long as the manuscript is not hypotheses-driven, but rather aimed at providing a direct analyses and comparison of two long-term datasets, the limitations to the observations at hand should be more carefully and clearly addressed.

We completely agree with this comment by Reviewer #1. Following his/her remark, we modified text in abstract as follows: "*In the Laptev Sea, **based on our data and methodology, DVM could not be detected** during polar night*". We also revised text in section 3.2 (first sentence of the third paragraph) as follows: "*In summary, **our data from** the Laptev Sea **does not indicate DVM** to observe during the major portion of polar night. **However, small-scale DVM (e.g., Ludvigsen et al, 2018) or DVM by smaller organisms than detected by a 300 kHz ADCP (<1.5 mm) cannot be fully discriminated**".* The acoustic backscatter data was obtained with a 30 min and 20 min ensemble time interval and 30 and 20 pings per ensemble for the Laptev and Beaufort Seas, respectively. It seems that acoustic backscatter from a single fish, if detected, is eliminated by averaging throughout the ensemble.

5. Section 3.3. and light data. It is not fully clear for me if figure 10 is based on actual measurements, or if they are modelled? If they are measured, I do not see a description of this in the methods? And if they are modelled, why not also include the polar night? There are available measurements and models that can be used as a source? Also, why use lux? Most other comparable studies use absolute quanta? I would at least aid comparison with other papers by making a rough estimate as to what the thresholds stated would be in Epar.

Figure 10 and all other figures showing irradiance based on modeling. This statement is in section 2.3, second paragraph: "*To quantify the total sky illumination at each site we used the skylight.m function from the astronomy MATLAB package (Ofek, 2014). The irradiance values were estimated at the ice-free surface or under the ice. Under-ice illumination was modeled using the exponential decay radiative transfer model (Grenfell and Maykut, 1977; Perovich, 1996)*". Under-ice irradiance was modeled throughout the entire period of observations below the ice also including the polar night. Figures 4-7b show actograms of modeled irradiance, and Figure 10 (left) shows modeled irradiance only for the portion of observations centered at winter solstices. To clarify this statement, we added "***modeled***" to figure captions of Figures 4-7b, 9, 10, 11, and 14a, d, g, and j. We also added "***modeled***" to description of actograms in section 2.3: "*Time-series of MVBS (Figures 4–7c–g) and surface layer **modeled** irradiance (Figures 4–7b), computed from the PIOMAS and ADCP estimates of sea-ice thickness…*". We agree with Reviewer #1 that some other comparable studies use absolute quanta, e.g. micromoles of quanta per second per square meter ($\mu mol\ s^{-1}\ m^{-2}$). To make our data comparable to that using absolute quanta we included both lux and $\mu mol\ s^{-1}\ m^{-2}$ to the color code at the bottom of Figures 4-7.

6. A general question: The site in the Laptev Sea is very shallow compared to the Beaufort Sea. This of course reflect a real difference, but it might also introduce a problem in comparing the two datasets, especially as the upward looking ADCPs are located very close to the sea floor: In periods with open water, can increased turbulence affect the measurements of DVM behaviour and/or the DVM behaviour itself? This is relevant for the discussion in section 4.3, but I do think that the perspectives of this discussion can be broader.

In shallow environment, storms during open water period result in sediment resuspension. The first paragraph of section 4.3 describes two storms at Anabar resulted in disruption of the DVM acoustic signal. At Khatanga, DVM signal was becoming noisier, but still traceable. Based on our data and methodology we cannot insist that storms stop DVM because the acoustic signal was becoming too noisier to trace DVM. Another problem is attributed to acoustic noise generated by riverine suspended sediments (also described in section 4.3) with similar circumstances for DVM. But in a case of riverine

suspended sediments, they only mask the acoustic signal generated by DVM making DVM hardly traceable.

Congratulations on a well written and important new paper!

Thank you.

**Reviewer #2**

We highly appreciate helpful comments and suggestions by Reviewer #1.

I commend the authors for a well written manuscript and interesting study. The manuscript uses mooring data from 2 contrasting polynyas in the Arctic Ocean. Key findings include the identification of different light intensity thresholds required to trigger DVM in the two seas, with the Beaufort Sea showing DVM even during polar night. The authors highlight the ecological implications of sea ice decline, emphasizing how polynyas disrupt synchronized zooplankton migrations.

The research is comprehensive, using robust data sets and providing valuable insights into the seasonal and spatial variability of DVM in polar regions. It underscores the importance of sea ice in regulating Arctic ecosystems and offers a nuanced understanding of the impacts of polynyas.

I recommend this paper for publication and have only some minor suggestions for improvement:

1. Line 165-170: Short or move to introduction

We moved this portion of text to the last paragraph of introduction.

2. Section 2.1: much of this reads like the introduction. I recommend to consolidated this sections and to move parts of the text to the introduction where the polynyas are already discussed

Originally, section 2.1 was placed to introduction. However, the majority of authors voted to move this section to Materials and Methods to keep Introduction strait forward. This is a reason to keep this text in section 2. Following this comment by Reviewer #2, we modify the title of section 2 as follows: "***Study Area,*** *Materials and Methods*".

3. Line 189: Add information that ADCPs were not calibrated, so volume backscatter data contains only relative presence-absence data

We added this information in description of ADCP observations as follows: "***Calibration of ADCP acoustic backscatter data was not conducted, so the volume backscatter represents only relative measurements***".

4. Line 277: rephrase sentence

We modified this sentence as follows: "*Thereby, dedicated image-processing strategies **allow to filter out unidentified cloud-artefacts by using multiple lead metrics in a fuzzy logic approach**…*"

5. Line 580: rephrase sentence

We rephrased this sentence as follows: "*At Khatanga, **correlation between MVBS and light was completely disrupted during** the last decade of March 2008 **following** consecutive events of polynya opening 5–20 km to the station (Figures 2, 12, 14k, and 14l)*".

6. Line 656: rephrase "plum off"

We changed this to "*Mackenzie River **water away from** the Amundsen Gulf...*".

7. Line 686: rephrase "discriminated"

We changed "*discriminated*" to "***neglected***"

8. Figure 1: Hard to read red text on some of the maps

We changed red to deep blue

9. Figure 4 - 7: Amazing figures!

Thank you!